# Post-Hurricane Damage Severity Classification at the Individual Tree Level Using Terrestrial Laser Scanning and Deep Learning

Carine Klauberg [1,*], Jason Vogel [1], Ricardo Dalagnol [2,3], Matheus Pinheiro Ferreira [4], Caio Hamamura [5], Eben Broadbent [1] and Carlos Alberto Silva [1]

1   School of Forest, Fisheries, and Geomatics Sciences, University of Florida, Gainesville, FL 32611, USA
2   Center for Tropical Research, Institute of the Environment and Sustainability, University of California Los Angeles (UCLA), Los Angeles, CA 90095, USA
3   NASA-Jet Propulsion Laboratory, California Institute of Technology, Pasadena, CA 91109, USA
4   Cartographic Engineering Department, Military Institute of Engineering (IME), Praça Gen, Tibúrcio 80, Rio de Janeiro 22290-270, RJ, Brazil
5   Federal Institute of Education, Science and Technology of São Paulo, Avenida Doutor Ênio Pires de Camargo, Capivari 13365-010, SP, Brazil
*   Correspondence: carine.klaubergs@ufl.edu; Tel.: +1-(352)-294-6885

**Abstract:** Natural disturbances like hurricanes can cause extensive disorder in forest structure, composition, and succession. Consequently, ecological, social, and economic alterations may occur. Terrestrial laser scanning (TLS) and deep learning have been used for estimating forest attributes with high accuracy, but to date, no study has combined both TLS and deep learning for assessing the impact of hurricane disturbance at the individual tree level. Here, we aim to assess the capability of TLS and convolutional neural networks (CNNs) combined for classifying post-Hurricane Michael damage severity at the individual tree level in a pine-dominated forest ecosystem in the Florida Panhandle, Southern U.S. We assessed the combined impact of using either binary-color or multicolored-by-height TLS-derived 2D images along with six CNN architectures (Densenet201, EfficientNet_b7, Inception_v3, Res-net152v2, VGG16, and a simple CNN). The confusion matrices used for assessing the overall accuracy were symmetric in all six CNNs and 2D image variants tested with overall accuracy ranging from 73% to 92%. We found higher F-1 scores when classifying trees with damage severity varying from extremely leaning, trunk snapped, stem breakage, and uprooted compared to trees that were undamaged or slightly leaning (<45°). Moreover, we found higher accuracies when using VGG16 combined with multicolored-by-height TLS-derived 2D images compared with other methods. Our findings demonstrate the high capability of combining TLS with CNNs for classifying post-hurricane damage severity at the individual tree level in pine forest ecosystems. As part of this work, we developed a new open-source R package (rTLsDeep) and implemented all methods tested herein. We hope that the promising results and the rTLsDeep R package developed in this study for classifying post-hurricane damage severity at the individual tree level will stimulate further research and applications not just in pine forests but in other forest types in hurricane-prone regions.

**Keywords:** 3D point cloud; artificial intelligence; conifer forest; disturbance; lidar





## 1. Introduction

The forest is an ecological and dynamic system influenced not only by topography, geographic location, and anthropogenic disturbances, but also by natural disturbances, such as those caused by windstorms. Disturbance affects the forest structure and the composition of species [1–3], and in this way has an effect on biodiversity and plant regeneration by increasing the light intensity in the understory [4], as well as altering water and carbon cycles [5]. These natural disturbances and their consequences are essential components of forest dynamics and are important for a forest's health and development. However, frequent disturbance episodes caused by tropical storm events may cause negative outcomes

where the forest cannot recover from the consequent damage. The resilience of the forest may be affected by extreme and constant disturbance, subsequent insect outbreaks, invasive species, intense wildfire due to the woody debris [6–10], alteration of forest demography, and tree injuries that reduce harvested timber value [11,12].

Hurricanes are one of the major natural disturbances to forest ecosystems in the Southeastern United States (US). A severe hurricane can extensively influence forest structure, composition, and succession [13], and can consequently cause related ecological, social, and economic damage [14]. Hurricanes with winds classified as Category 4 (119–253 km/h) or 5 (>253 km/h) generally have 'catastrophic' effects on forests [15]. On 10 October 2018, Hurricane Michael caused landfall in the panhandle of Florida as a Category 5 hurricane [16] and caused an estimated $15 billion in damages, including more than $5.18 billion in losses to the agriculture and timber industries. In Florida alone, Hurricane Michael damaged more than 1.1 M ha of forest, with over 560,000 ha severely or catastrophically damaged (>75% of the forest downed). In the Florida Panhandle, due to extraordinary winds, Hurricane Michael led to a recorded tree mortality rate as high as 80% in some areas, and the single event affected 28% of all extant longleaf pine forests [17]. The storm continued inland, causing a tree mortality rate exceeding 20% as far as 150 km inland, where it continued to be classified as a Category 2 storm [16] and caused catastrophic effects in forests in Georgia and Alabama. Hurricane Michael changed the overall structure of the present forests, modifying the forest canopies by creating large gaps and downed trees where it passed [18].

High-frequency and -intensity hurricane events can create different types and amounts of tree damage, potentially affecting pathogen outbreaks, fire regimes, and the subsequent availability of seed sources for regeneration [12,19]. Trees that are twisted or bent by strong winds could lose vascular connection to root systems, making them susceptible to insect or pathogen attacks. Fire regimes could be altered if downed trees are lying entirely on the ground where higher moisture contents would reduce fire intensity, or if they are 'hung up' and elevated aboveground in some manner where tissues would more easily dry out. In the latter case, hanging tree crowns could facilitate ground fires climbing into the canopy, where the resulting fires are more difficult to control and more likely to kill the mature remaining live trees. Tree mortality from either wind or any resulting factors (insects/pathogens, fire) could affect tree regeneration, especially for species, like longleaf pine (*Pinus palustris* Mill.), that mast or infrequently produce seeds. The devastated area in the Florida Panhandle includes a longleaf pine habitat that is classified as a global biodiversity hotspot and endangered [14,15]. Although longleaf pine forests have, for over millennia, been exposed to frequent storm events and wildfires, making them resistant to these disturbances [20,21], maintaining longleaf pine ecosystems requires a significant amount of management.

Measuring forest damage and other factors, such as carbon storage, tree diseases, and fuel amount, provides a basic quantification of a forest's potential scenarios and consequent forest management practices [22]. The most accurate way to evaluate forest damage is by doing forest inventory in situ as a method to conduct a quality and quantitative diagnosis as a guide to forest management. The forest attribute data may be collected by a trained team using ground assessments. However, forest inventory is expensive and requires a lot of time [23], and it is not possible to assess each tree and access all landscapes that will be evaluated, particularly post-hurricane. Remote sensing has proven to have a high potential and capacity to map and estimate forest attributes, e.g., species structure, canopy height, carbon and fuel amount, etc. [24].

Remote sensing tools and techniques offer the possibility to monitor and assess the impacts of forest disturbances like windstorms at the landscape level. One of the first and most notorious studies using remote sensing to assess hurricane damage was the study from [25], which used Landsat and MODIS optical data to assess the damage from Hurricane Katrina, which affected the US Gulf Coast back in 2005 [25]. Besides using the remote sensing data, they also inventoried the affected areas to train their models. The

parameters of vegetation structure are arduous and onerous to measure in the field, and it is a challenging task, especially after a hurricane, for example. More recently, light detecting and ranging (lidar) data from terrestrial, airborne, or spaceborne systems have proven to be useful in mapping forest structure and canopy cover [26], determining forest health [27], stem detection [28], tree diameter [29], and biomass [30], and extracting tree variables such as height [31] and crown diameter [32]. The importance of the terrestrial lidar system, known as terrestrial laser scanning (TLS), has grown in view of its ability to provide 3D point cloud data with high precision, the fact that it is relatively easy to interpret, and the fact that parameter extraction can be automated [33–35]. However, most of the previous research has been conducted on stands and single trees [36], and none of the research relates to the damage to single trees after a natural disturbance by hurricanes.

In recent years, deep learning methods have become important in remote sensing [37] and are gaining popularity in the forestry field [38–40]. For instance, Ferreira et al. [41] proposed the species classification of Amazonian palms using convolutional neural networks (CNNs) in UAV images. CNNs in an input image (e.g., acquired from UAV) automatically perform feature extraction of target objects (e.g., trees, buildings, etc.) and, based on specific features, they can classify objects within the input image (e.g., tree species) [37]. Nezami et al. [42] used hyperspectral and RGB imagery with CNNs to facilitate tree species classification. Implementing deep learning to 3D data from lidar brings unique challenges and could provide the highest performance development in classifying image components [43]. Directly using the lidar points often requires prohibitive effort, as shown by Xi et al. [44], who assessed the effectiveness of machine learning and deep learning for wood filtering and tree species classification from TLS. In the case of airborne lidar data, one possibility is simply to use the canopy height model (CHM), therefore an image showing the top of the crowns, as has been done in a study segmenting palm trees across the Brazilian Amazon forests using a U-Net model [40]. Another way of working with the point data is through a methodology named SimpleView, which takes snapshots of the 3D trees over different viewpoints and uses those to train deep learning models [43]. This method has been recently used to identify tree species with TLS data and a ResNet model [45]. In general, classification from image or scan datasets is still a challenge due to the uncertainty over classifier selection.

TLS and deep learning combined could provide an efficient way of assessing post-hurricane damage severity at the individual tree level. For instance, deep learning algorithms can be used to analyze images derived from TLS data, while TLS data can be used to provide information about forest structure, allowing the deep learning algorithm to accurately classify damage severity. Therefore, the aim of this study was to assess the capability of TLS and CNNs to classify post-Hurricane Michael damage severity at the individual tree level in a pine-dominated forest ecosystem in the Florida Panhandle, Southern U.S. More specifically, we assessed the combined impact of using either binary-color or multicolored-by-height TLS-derived 2D images along with six CNN architectures for post-hurricane damage severity classification at the individual tree level, as well as to support and enhance TLS-based forest inventory, monitoring, and conservation initiatives. As part of this study, we developed rTLsDeep, a new open-source R package for post-hurricane individual tree-level damage severity classification using TLS and CNN architectures [46].

## 2. Materials and Methods

### 2.1. Study Area

Forests of the Southeastern U.S. are highly biodiverse (e.g., the longleaf pine (*Pinus palustris* Mill.) ecosystem), and productive (e.g., Loblolly pine (*Pinus taeda* L.) forest plantations), and natural disturbances by hurricanes are well-known to be an important source of mortality. For this study, we focused on the northwestern panhandle area in Florida, near Apalachicola National Forest—ANF (30°19′15.74″N, 84°52′0.02″W), where Hurricane Michael caused severe to moderate destruction in November 2018 (Figure 1). This region includes the national forest, the nearby state forest, and some private forests that are man-

aged for timber, recreation, and/or wildlife purposes. The United States Forest Service (USFS) has a network of permanently fixed area plots as part of a RESTORE Act-funded project with objectives including restoring hydrologic function in the zone. Further, this region was classified as a "hotspot" in North America, and threatened and endangered species were observed and studied. The topography is relatively flat, and either the Gulf of Mexico or the Apalachicola River receives the water from existing small streams in large quantities. The predominant tree species are pines (*Pinus* spp.), though hardwoods and bald cypress (*Taxodium distichum* (L.) Rich.) can be found. A highly diverse understory of grasses, forbs, and shrubs is found in pine forests, where low-intensity fires are common. The majority of fauna and flora species in this zone are adapted to fire, and prescribed fires are used for fuel management as a way to reduce a wildfire's negative impacts and to promote the development of desired understory plants. The mean elevation is 5 m, while the mean annual precipitation is 1395.8 mm. The average temperature annually ranges from 15 to 25 °C.

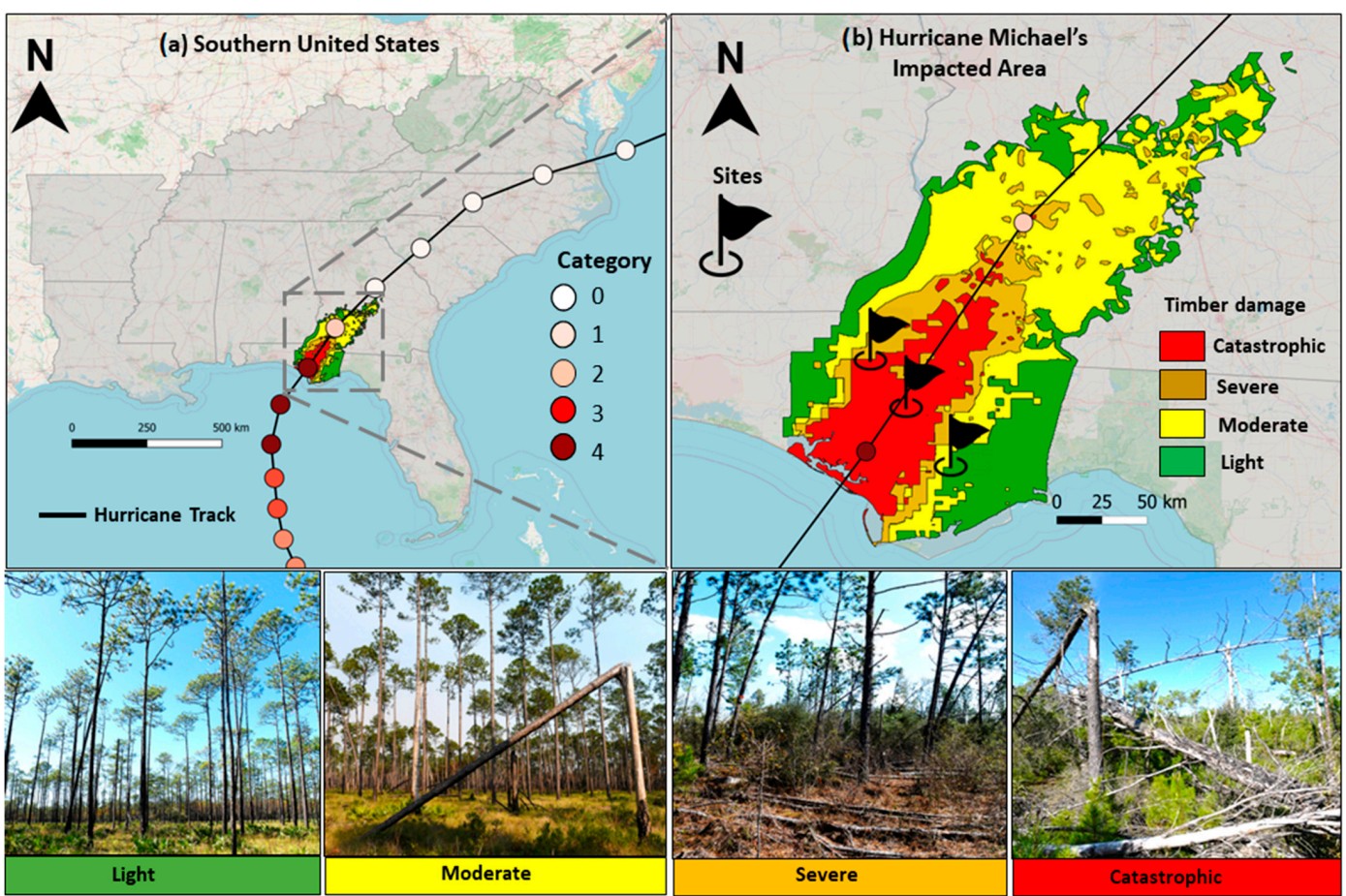

**Figure 1.** Hurricane Michael path and category in the Southern US and study sites located within the area impacted by Michael within the Apalachicola National Forest (ANF) and private forests areas (**a**). Area impacted by Hurricane Michael (36,218.00 km²) and study sites location (**b**). The timber damage severity map was produced by the Georgia Forestry Commission and Florida Forest Service [47].

### 2.2. TLS Data Acquisition and Processing

TLS data were obtained in November 2021 across longleaf pine and sand pine (*Pinus clausa* (Chapm. ex Engelm.)) forest stands within public and private lands in the Panhandle area affected by Hurricane Michael. The sites were pre-selected after the disturbance event based on a visual assessment of pre- and post-disturbance Google Earth aerial images. In the field, we visited the pre-selected areas and selected the final sites, covering the entire range of damage severity (light, moderate, severe, and catastrophic; see Figure 1). Within

12 16.92-m fixed radius plots (900 m$^2$) distributed in three sites, five TLS scans (four in the edge—north, south, east, and west; one in the center) were obtained using Riegl VZ 400 i coupled with a NIKON D850 45.7 MegaPixel digital camera and a differential GNSS RTK Receiver. The scan configuration was set to panorama 20 and frequency 1200 Hz. The point cloud pre-processing, including point cloud registration, noise removal, and clipping, was carried out using RiSCAN Pro® [48] (Figure 2a). Using CloudCompare® [49] a total of 90 individual trees were randomly and manually extracted from the point cloud with 15 trees per class of damage severity (Figure 2b): C1—no damaged tree (intact tree, no visual damage), C2—leaning tree, C3—beading tree with a trunk like a bow, C4—trunk snapped with stem or crown broken, C5—stem breakage (no crown), C6—fallen or uprooted tree.

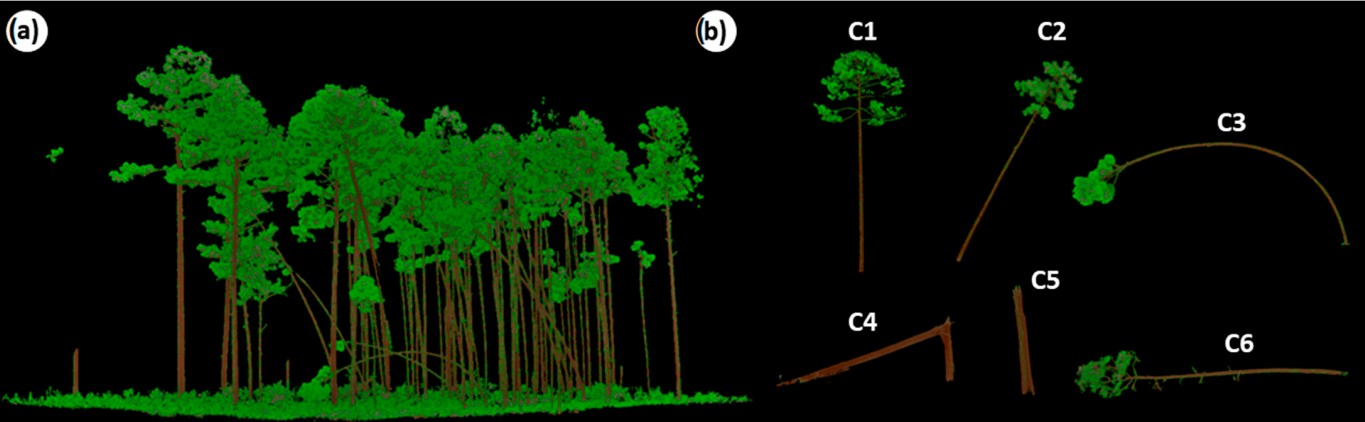

**Figure 2.** TLS-derived 3D point cloud at plot level (**a**) and extracted individual trees (**b**): C1—no damaged tree (intact tree, no visual damage), C2—leaning tree, C3—beading tree with a trunk like a bow, C4—trunk snapped with stem or crown broken, C5—stem breakage (no crown), C6—fallen or uprooted tree. The damage severity classification was adapted from previous studies (e.g., Rutedge et al. [15]).

Using the a priori damage severity classification, we derived 12 (1500 × 1500 pixels) 2D images from the 3D point cloud for every single individual tree extracted using the tlsrotate3d function from the rTLsDeep package in R developed in this study [46] (see Supplementary Material, Figures S1–S6). Each of the 12 images corresponds to a different viewing angle according to the rotation in the Z-axis at each 30° increment (from 30° to 360°) (Figure 3). We created two sets of 2D images, one using binary colors (black and white) and the other being multicolored by tree height (0–30 m) (Figure 3). In total, we created 1080 2D images per color class (in binary color and multicolored by height) that were then used as inputs in six CNN architectures (Figure 3).

*2.3. CNN Models and Accuracy Assessment*

In this study, we used six of the most used CNN architectures for damage severity classification at the tree level as follows:

(i) Densenet201, a densely connected convolutional network (DenseNet), was proposed by Huang et al. [50] and introduced a novel framework to connect the layers of a CNN. In a DenseNet, each layer takes all preceding feature maps as input and passes its own feature maps to all subsequent layers. Here, we used a DenseNet architecture that is 201 layers deep and has been widely used in vegetation remote sensing [51].

(ii) EfficientNet_b7 [52]—EfficientNets are a family of CNNs that have achieved outstanding performances with a reduced number of parameters. They were introduced by Tan et al. [52] and rely on the so-called compound coefficient that uniformly scales all dimensions (depth, width, and resolution) of the network. EfficientNet_b7, used in this work, has a compound coefficient that equals 7 and achieved the best results among all EfficientNet variants tested by [52].

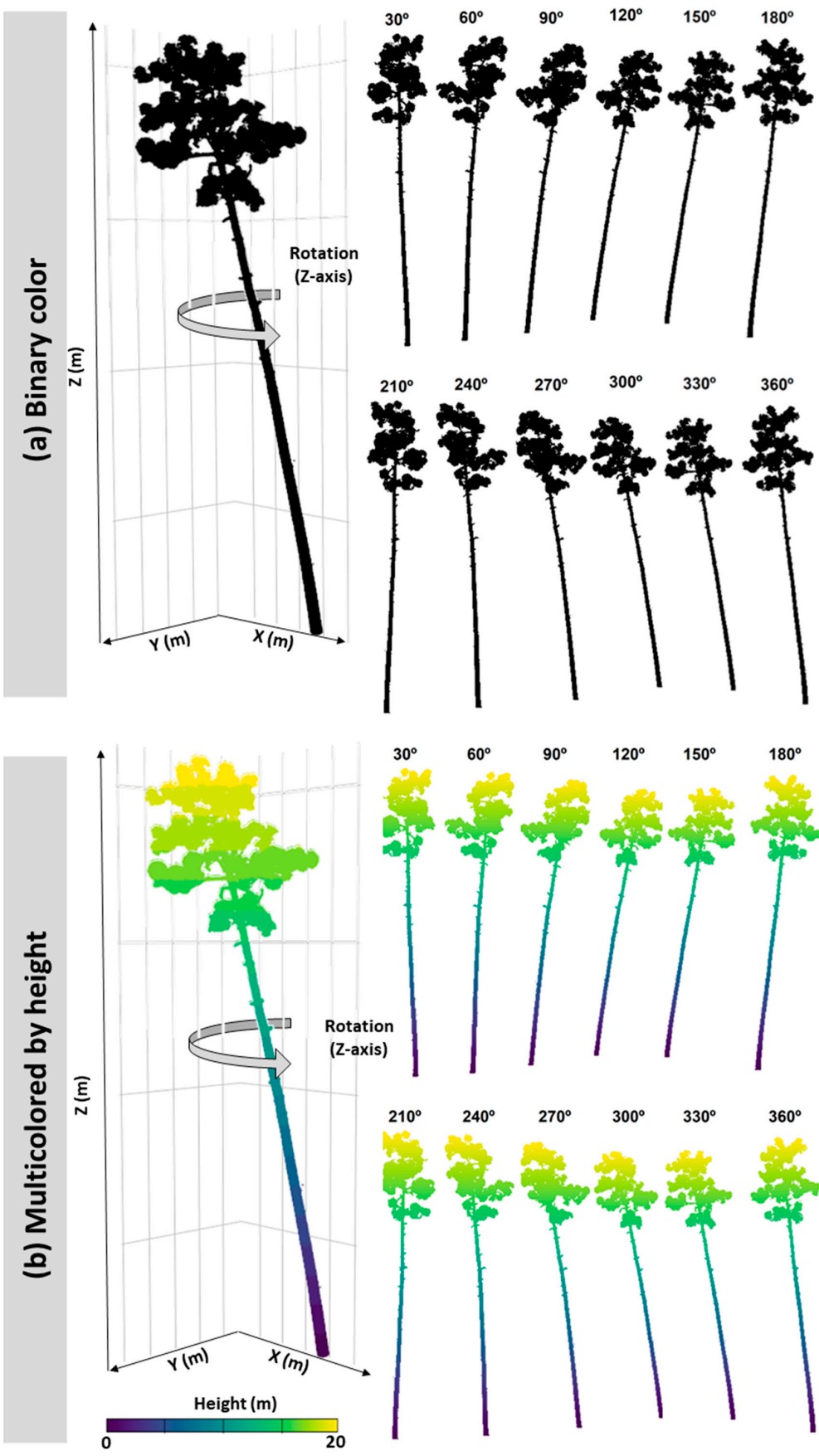

**Figure 3.** An example of tree-level 3D point cloud rotation (Z-axis by 30°) and 2D image computation as (**a**,**b**).

(iii) Inception_v3 [53]—The Inception architecture was first proposed by Szegedy et al. [53,54] in a network called GoogLeNet. It allowed GoogLeNet to have a reduced set of parameters (e.g., 12 times fewer parameters than AlexNet [55]) and still provide outstanding classification results. However, its architecture and design are complex and changes to the network are prohibitive and often hamper computational gains. Szegedy et al. [54] proposed a set of modifications to the Inception architecture, such as factorizing convolutions with large filter sizes, factorization into smaller convolutions, and spatial factorization into asymmetric convolutions. These modifications allowed a significant improvement in classification accuracy while maintaining network complexity and keeping computational costs low [53,54].

(iv) Resnet152v2 [56]—Previous research on CNNs (e.g., [57,58]) showed that increasing the number of layers (the depth) of a CNN can improve feature extraction and, consequently, classification accuracy. However, the experimental analysis showed that an increase in the network depth leads to an increase in the training error [58]. He et al. [59] proposed a novel framework that allowed training models with many layers, thus improving feature extraction while maintaining the trade-off between classification accuracy and computational cost. This framework is based on residual blocks with skip connections that forward the feature maps of a given layer to a deeper layer in the network, giving rise to the residual network (ResNet) family. Many ResNet variants exist that differ from each other by the number of residual layers. Here, we used the variant Resnet152v2 [56], which is composed of 152 layers with identity mappings as skip connections. Identity mappings reduce the difficulty of network convergence by transferring to deep layers feature maps from shallow layers.

(v) VGG16 [58]—The VGG16 model, proposed by Simonyan et al. [58], was one of the first networks to overcome AlexNet [55] in the large-scale visual recognition challenge (ILSVRC), a renowned international competition that evaluates algorithms for object detection and image classification. VGG16 is a simple model composed of only 16 layers.

(vi) A simple CNN variant composed of two convolutional layers and one max pooling layer. We have laid out our approach in Figure 4. First, an input image was passed through a set of convolutions, pooling, and fully connected layers for feature extraction (Figure 4b). Then, the softmax classifier was applied to retrieve class membership probabilities, and the input image was classified according to the class that achieved the highest probability score (Figure 4c). The parameters of all networks were initialized with pre-trained values of the ImageNet database [60] except the simple CNN variant, which was trained from scratch. During training, to update the network hyperparameters (weights and biases), we used the adaptive moment estimation (ADAM) optimizer [61].

We have implemented all CNN models in our rTLsDeep package [42]. From the total of 1080 TLS-derived 2D images per color class (in binary color and multicolored by height), 80% of them (*n* = 864) were used for training, and the remaining 20% (*n* = 216) were used for testing. During the construction of the training and testing sets, the tree identity was respected. This means that if a given tree was selected for training, all of its 12 images (corresponding to different viewing angles) were used to train the model and not used for testing. For assessing the classification accuracy, we computed the overall accuracy (OA), F1-score, and the Kappa index (Equations (1)–(3)). Moreover, we computed the confusion matrices for each model. The diagonal cells of a confusion matrix show the amount of correctly classified samples, while the off-diagonal cells show the misclassification rate.

$$\text{OA} = (\text{TP} + \text{TN})/(\text{TP} + \text{TN} + \text{FP} + \text{FN}) \tag{1}$$

$$\text{F1-score} = (\text{TP})/(\text{TP} + \frac{1}{2} \times (\text{FP} + \text{FN})) \tag{2}$$

$$\text{Kappa} = (po - pe)/(1 - pe) \tag{3}$$

where TP is the true positive, FP is the false positive, FN is the false negative, TN is the true negative, and ns is the total number of samples. *po* is the proportion of trees correctly classified and *pe* is the expected proportion of trees correctly classified by chance [62].

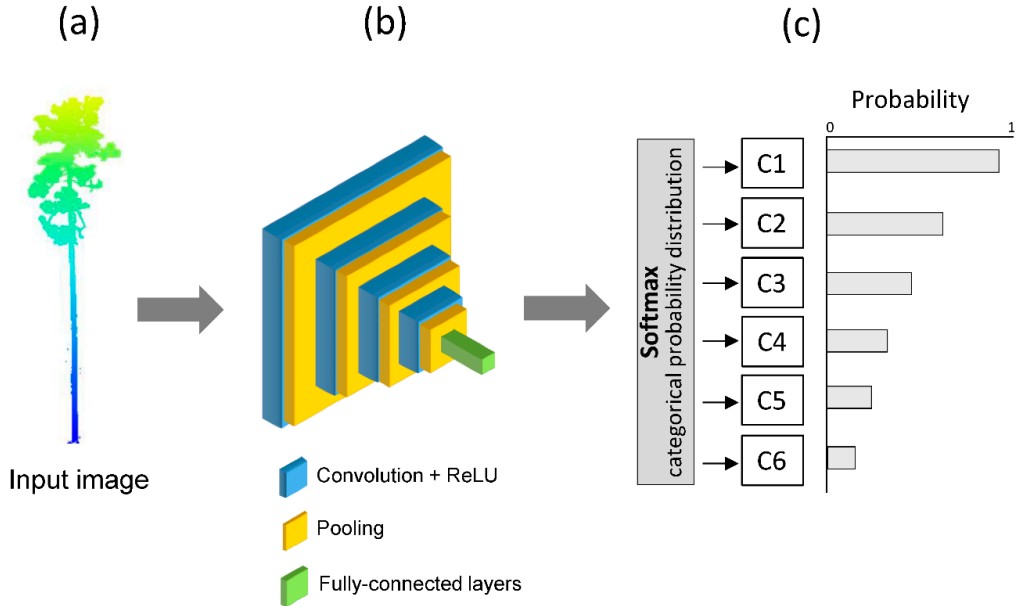

**Figure 4.** Damage severity classification approach using convolutional neural networks (CNNs). The input image (**a**) passes through a set of convolutions, pooling, and fully-connected layers (**b**) that perform feature extraction. At the end of the process (**c**), the softmax classifier is applied to retrieve class membership probabilities. The input image is then classified according to the class that achieved the highest probability.

## 3. Results

Based on the six deep learning architectures, we had a mean overall accuracy of 87.34% for the binary-colored images and 86.59% for the images multicolored by height for classifying damage level severity at the tree level from TLS data (Table 1). In general, all CNN classifiers had similar Kappa statistics and mean F1-score values, except in the simple CNN architecture with multicolored-by-height 2D images, whose values were 20% less than general (Table 1). Considering the overall accuracy for binary-color images, VGG16 attained a slightly higher accuracy compared to the other CNN architectures. For the multicolored-by-height images, VGG16, Inception_v3, and ResNet152v2 showed higher overall accuracy, F1-scores, and Kappa values compared to other CNN architectures.

A detailed F1-score by individual classes reveals a higher accuracy for predicting damage severity in classes C3 to C6 compared to the accuracy in classes C1 and C2 (Figure 5). In general, when using images multicolored by height, CNN architectures show better performance based on the F1-score. VGG16 and EfficientNet_b7 had superior F1-score values in all six CNN architectures when using binary color images. For the multicolored-by-height images, VGG16 and Inception_v3 had the highest F1-score in all classes assessed (Figure 5).

**Table 1.** Summary of overall accuracy, kappa statistics, and mean F1-score for six models' approaches within image type (colored by height and black) from the validation dataset. The best results by image type are shown in bold.

| Validation Data | Architecture | Overall Accuracy (OA) | Kappa Statistic | Mean F1-Score |
|---|---|---|---|---|
| **Binary color (Black and white)** | DenseNet201 | 0.8725 | 0.84631 | 0.87107 |
| | EfficientNet_b7 | 0.8435 | 0.85814 | 0.88065 |
| | Inception_v3 | 0.8774 | 0.85228 | 0.87754 |
| | ResNet152v2 | 0.8725 | 0.84631 | 0.87027 |
| | Simple CNN | 0.8823 | 0.85828 | 0.88307 |
| | VGG16 | **0.8922** | **0.86995** | **0.89321** |
| **Multicolored by height** | DenseNet201 | 0.8725 | 0.84637 | 0.87253 |
| | EfficientNet_b7 | 0.8627 | 0.83403 | 0.84620 |
| | Inception_v3 | 0.9020 | 0.88166 | 0.89777 |
| | ResNet152v2 | 0.9012 | 0.88182 | 0.90512 |
| | Simple CNN | 0.7353 | 0.67936 | 0.70929 |
| | VGG16 | **0.9216** | **0.90556** | **0.92337** |

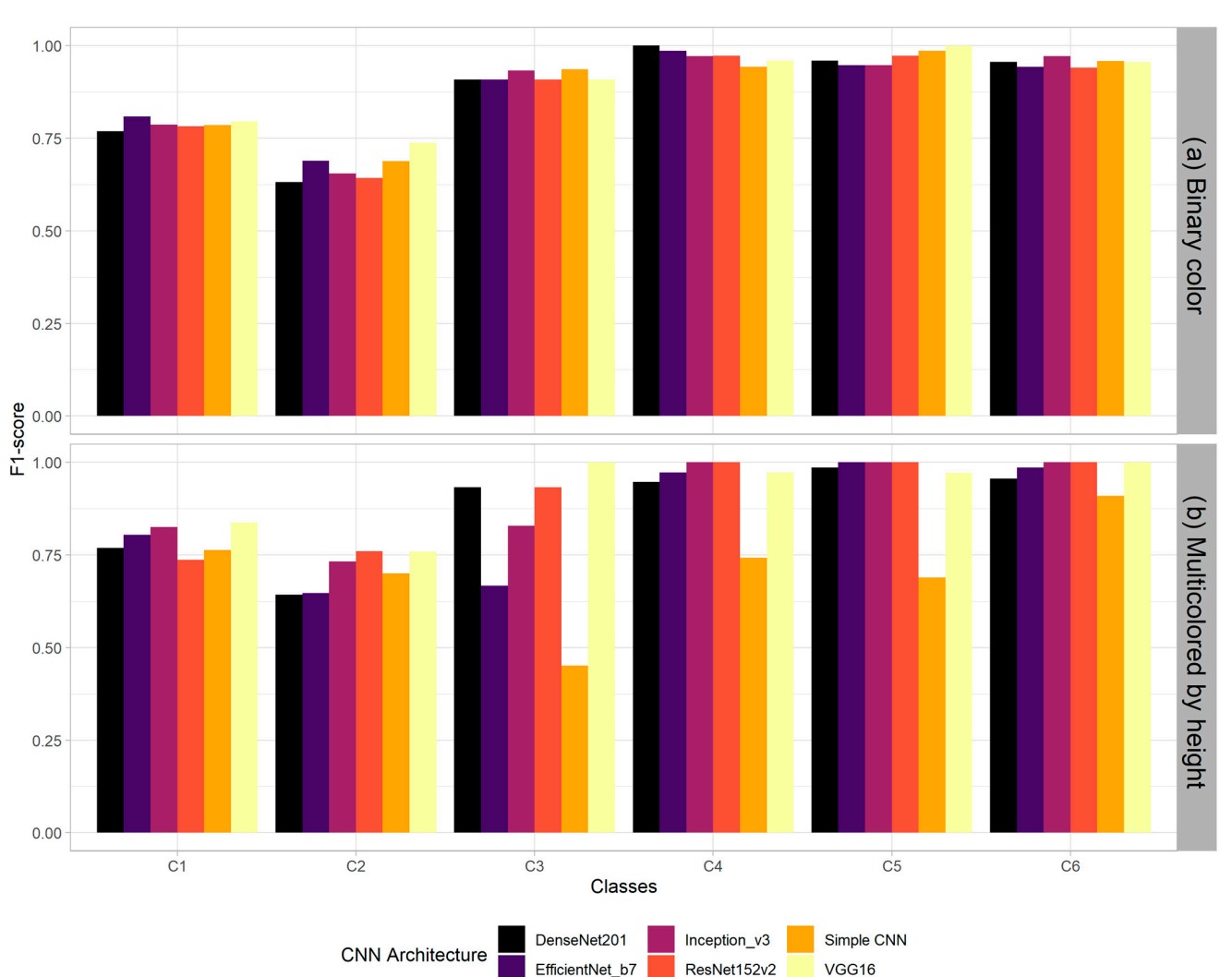

**Figure 5.** F1 score for each CNN architecture across post-hurricane damage severity categories (**a**,**b**).

Based on the confusion matrices (Figure 6), both within the binary-color and multicolored-by-height 2D images, we found many samples that were mistakenly classified in classes C1 and C2. Commonly, C1 is misunderstood as C2 and C3. The trees in C2 are also confused with trees in C1 and C3. The confusion matrices show that all models had accurately classified damage severity levels in C3 (95.7% to 100%), C4 (92.3% to 100%), C5 (90% to 100%), and C6 (94.6% to 100%) when using binary-color images and in C3 (100%), C4 (59.0% to 100%), C5 (90.9% to 100%), and C6 (100% correct) when using multicolored-by-height images. For the best overall CNNs, VGG16, Inception_v3, and DenseNet201 were the most frequently accurate. The simple CNN variant, compared with all the other CNN architectures, showed the lowest accuracies on both sets of 2d images.

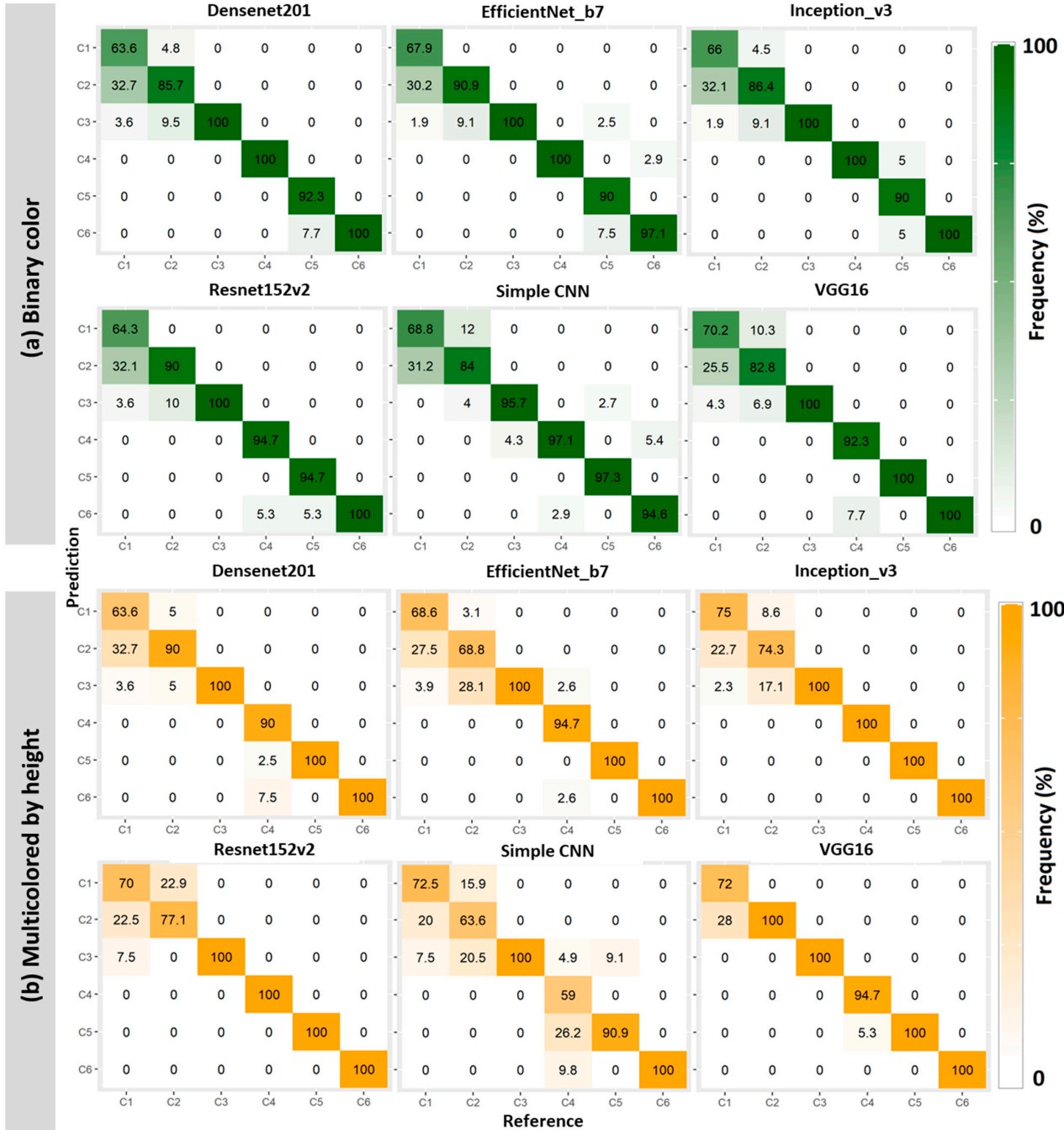

**Figure 6.** Confusion matrices derived from CNN architectures and TLS-derived 2D images in (**a**,**b**). The percentage of misclassification between a pairwise combination of damaged trees is shown in the off-diagonal cells. Each cell contains absolute values (pixels) and relative percentages.

## 4. Discussion

We tested six CNN architectures for classifying post-hurricane damage severity at the individual tree level using 2D images derived from 3D point clouds acquired by TLS. Previous research demonstrated the potential of lidar data and deep learning methods for leaf and wood separation and tree species classification [14,40,44]. However, as far as we know, this is the first study demonstrating the potential of combining TLS with deep learning models for classifying damage severity at the tree level after a hurricane disturbance. The results showed that CNN architectures in our study had satisfactory performances. An accurate estimation of the damage severity after a major disturbance event is very important for prioritizing forest management practices. For land managers, damage severity classification can be used in response to future hurricanes, for economic timber evaluation, and to manage potential wildfires. In this section, we discuss the results and highlight future research needs.

The TLS dataset is from an environment with a stressor, which induces significant changes to the tree structure. The transformation of 3D point clouds to images allowed the usage of powerful and established techniques of image classification based on CNNs, while the direct use of 3D point clouds would have been computationally onerous [63]. The summary of the accuracy results from all the architectures tested in our approach was promising (overall accuracies ranging from 67.9% to 92.3%). Although overall accuracy can indicate performance, it could be, to a certain degree, inaccurate in a situation where there is an unbalanced number of samples in the training dataset [42]. Hence, the F-1 score and Kappa value were also used to assess damaged trees by class and by the type of model that was tested.

Our results show that damaged trees in classes C3, C4, C5, and C6 could be classified with high accuracy rates (F1-score > 0.94). When comparing the confusion matrices, we can see that classes C3, C4, C5, and C6 were accurately classified. This classification success is attributed to the distinct architecture of these four classes. The shape of C4 looks like a triangle, C5 has no crown, and C6 is horizontally arranged. In contrast, C1, C2, and C3 have in common the presence of trunks and crowns, and the inclination of the whole tree is used as a parameter for dissimilarity among them. The image representation of the point clouds multicolored by height is also more distinguished in classes C4, C5, and C6 compared to in C1, C2, and C3. C1 and C2 used all established ranks of colors in most of the samples, which is the opposite scenario in classes C4, C5, and C6 as these trees are not tall as in C1 and C2. For example, C6 is horizontally laid out and will not achieve a mean height value. The capability of CNNs models to learn the tree architecture characteristics in 2D images is inspired by the human visual cortex [64], and both CNNs and human vision are susceptible to errors. For example, we observed relatively high rates of misclassification among C1, C2, and C3 (C1 to C2 ~28% and C2 to C3 ~11%). The classification errors among these classes may have resulted from a different point of view depending on the rotation in the x-y plane. For instance, the C2 trees might have been misclassified as C1 because the trunk inclination of this is less than 45°, and in a different position it could have appeared to be a straight trunk. However, trees with no damage (C1) and leaning (<45°) (C2) stems are feasible for forest harvest since these trees can still be merchandised as sawlogs [16].

Regarding the performance of all the CNN architectures tested in our study, the VGG16, DenseNet201, and Inception_v3 approaches yielded results with less confusion and high accuracy values than the other methods. The simple CNN model showed an overall weaker performance in three out of the six tested damaged tree classes. One potential explanation for the difference in accuracy between the simple CNN and the remaining models is that all the other models were pre-trained with ImageNet data and then fine-tuned with the labeled training data, while the simple CNN relied entirely on our small training data to adjust the model weights. Therefore, all models but the simple CNN leveraged transfer learning, that is, the previous weights that were trained with other images helped to better identify the tree damage classes. A direct and specific cause of the difference in prediction accuracy associated with the six CNN models is complex to

determine, and many factors might have impacted the accuracy, including the selection of the training and test datasets [63], data resolution, and attribution [44]. Additionally, small samples (<40 trees) may explain the poor performance of CNN models, but limited sample sizes are not unusual for TLS research in many ecosystems [64]. Further, classification accuracy is not only affected by data resolution and sample size but also by the choice of classifiers. In methods of classification based on deep learning, convolution, and pooling layers will vary among the architectures and will affect these. The pooling can be used to remove anomalous pixels, and it is a process of down-sampling to obtain an average or maximum value in a near portion [65]. Deep learning classifiers, i.e., VGG, ResNet, and DenseNet, have different parameter sizes and each one will respond in accuracy, rapidness, and stability in a distinct way. As such, DenseNet was drawn to have a small parameter size and to be a weightless model [44].

In this study, the whole procedure of using plot scans to classify damage severity at the tree level was not totally computerized, and the segmentation of individual trees by damage severity class was executed manually. However, the general high accuracy rates found herein set out the advantages of our proposition—the application of deep learning and TLS data in disturbed areas—and they are an important first step in prompting an automated forest inventory for after hurricanes that would provide wood and carbon storage estimates, as well as support research based on the ecological process. In this way, a deep learning approach may be used in future applications due to the potential performance to solve segmentation problems. Also, the direct use of 3D point clouds in the damage severity classification procedure might be considered for improving performance accuracy and assisting with the quantification and qualification of the trees after a disturbance for forest management practices, however, the trade-off between accuracy and computation must be further investigated.

## 5. Conclusions

In this study, we assessed the capability of TLS and deep learning for classifying post-hurricane damage severity at the individual tree level in a pine-dominated forest ecosystem in the Florida Panhandle. Combining TLS with six types of CNNs was shown to be efficient for classifying post-hurricane damage severity at the individual tree level with a high accuracy. However, VGG16 and multicolored-by-height TLS-derived 2D images outperformed all the other methods tested. This is the first attempt to combine TLS and deep learning for classifying damage severity at the tree level. Despite the promising results found herein, there is still a long path to run until the proposed method can be applied at an operational scale. Improvements are needed not only in damage severity classification, but also in the efficiency and automation of methods for individual tree extraction, especially fallen trees, from TLS data. We hope the open-source rTLsDeep R package developed in this study for classifying post-hurricane damage severity at the individual tree level will stimulate further research and applications not just in longleaf pine but other forest types in hurricane-prone regions.

**Supplementary Materials:** The following supporting information can be downloaded at: https://www.mdpi.com/article/10.3390/rs15041165/s1, Description of the rTLsDeep R package (Figures S1–S6).

**Author Contributions:** Conceptualization, C.K., R.D., M.P.F. and C.A.S.; methodology, C.K., R.D., M.P.F. and C.A.S.; software, C.K., R.D., M.P.F., C.H. and C.A.S.; validation, C.K., R.D., M.P.F. and C.A.S.; formal analysis, C.K., R.D., M.P.F. and C.A.S.; investigation, C.K., R.D., M.P.F. and C.A.S.; resources, C.K., R.D., M.P.F., J.V., E.B. and C.A.S.; data curation, C.K., R.D., M.P.F. and C.A.S.; writing—original draft preparation, C.K., R.D., M.P.F. and C.A.S.; writing—review and editing, C.K., R.D., M.P.F., C.H., E.B., J.V. and C.A.S.; visualization, C.K. and C.A.S.; supervision, C.A.S., E.B. and J.V.; project administration, C.K.; funding acquisition, J.V. and E.B. All authors have read and agreed to the published version of the manuscript.

**Funding:** This research was funded by the USDA NIFA Awards # 2020-67030-30714 and #2023-68016-39039.

**Acknowledgments:** We would like to express our sincere thanks to the funding agency and Danilo R. F. Souza and Luiz A. Nogueira for assisting with the rTLsDeep package in R and Rodrigo Leite for the field data collection.

**Conflicts of Interest:** The authors declare no conflict of interest.

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
