# Peer review of "Post-Hurricane Damage Severity Classification at the Individual Tree Level Using Terrestrial Laser Scanning and Deep Learning"

_remotesensing, doi:10.3390/rs15041165_

Round 1

Reviewer 1 Report

Dear Authors,

I have reviewed the manuscript. The topic and contents are interesting. Some parts of the manuscript need to be improved such as Introduction and Methodology. You can find the required corrections in the attached PDF.

Author Response

Thank you for your comments and your reflections on our paper. We believe your contributions have offered great value to this manuscript and we are ecstatic with the opportunity to address the questions you have raised here. We are certain that now we have an even stronger manuscript, and we are extremely thankful for your important remarks.

Reviewer 2 Report

The manuscript under review focuses on an interesting topic that falls within the scope of Remote Sensing, as is the aim for jointly use terrestrial laser scanning and deep learning to predict tree damage severity following in pine forests follow a severe perturbation (hurricane Michael). The manuscript is quite innovative, and is quite well written and easy to follow. While I have in general a very favourable opinion on the manuscript, there are some few minor concerns that I consider should be solved by the authors before recommending publication. In the next paragraphs, I’ll expose this minor criticism:

Abstract

-          Lines 32-36: here, and also in line 200 (section 2.2) the authors make mantion to the rTLsDeep package, indicating that has been specifically developed for this study. If this is the case, more information about the characteristic should be provided in the text , or at least as supplementary material

Introduction

-          Lines 115-116: given that the target reader of Remote Sensing may not be familiar with deep learning methods, and taken into account that only CNNs are used, I suggest to include somewhere in this section a brief description (3-4 lines ad maximum) of what a convolutional neural network is and how it works

Material and methods

-          Line 155 and figure 1. How was the timber damage classification shown here carried out? In figure’s legend it indicates that the map was produced by Georgia and Florida forest services, but it is not clear the subjacent basis. By means of field survey, remote sensing, direct assignation by the intensity of the hurricane?

-          Line 182: Which are the characteristics of these three sites? Level of damage? ISs there any data on the pre-hurricane attributes of the forest in each site (age. Mean height, basal area, stand density)? How were these sites selected?

-          Line 182: How was selected the location of the 4 plots within each site? Were plot homogeneous within and between sites, or they aimed to cover different degree of initial conditions or damage severity?

-          Line 187-188: The authors should indicate which was the criteria for selecting the individual trees finally segmented from the cloud point. If only easy to see, free of competition trees, are subjectively selected, this could lead to some bias in the accuracy of the classification method that at least should be discussed in deeper detail

-          Line 189-190: This damage severity classification was specifically proposed for this work or it's been used previously ?In that case provide a reference

-          Line 202-204: change the numeration of the figures (currently figure 4 is presented in the text before figure 3)

-          Lines 214-216: I miss a short description of the characteristics of the six different CNN methods evaluated, focusing on their characteristics, and the difference between them (including topics as free access, easiness for using …). This information could be provided as a table, even as an appendix. Moreover, indicate why these methods were selected, and even, why among the deep learning methods, CNN were selected among others

-          Line 230: please provide description (formulae) of OD, F1-score and Kappa index

Results

-          Line 264: what do the authors mean by “the confussion matrices were symmetric”? Confussion matrices are not symmetric at all.

Discussion

-          Lines 318-320: This pre-training process has not been presented neither in methods section nor in results. Since this may be a real cause for the superiority of some CNN techniques over others this should be detailed in these sections.

-          Lines 347-350: without demerit at all of the quality and interest of the work I consider that the authors should be more realistic in indicating and discussing here that there is still a long path to run until the proposed technique may be used at an operational scale  

Author Response

The manuscript under review focuses on an interesting topic that falls within the scope of Remote Sensing, as is the aim for jointly use terrestrial laser scanning and deep learning to predict tree damage severity following in pine forests follow a severe perturbation (hurricane Michael). The manuscript is quite innovative, and is quite well written and easy to follow. While I have in general a very favorable opinion on the manuscript, there are some few minor concerns that I consider should be solved by the authors before recommending publication. In the next paragraphs, I’ll expose this minor criticism:

Authors: Thank you for your comments and your reflections on our paper. We believe your contributions have offered great value to this manuscript and we are ecstatic with the opportunity to address the questions you have raised here. We are certain that now we have an even stronger manuscript, and we are extremely thankful for your important remarks.

Abstract

-          Lines 32-36: here, and also in line 200 (section 2.2) the authors make mantion to the rTLsDeep package, indicating that has been specifically developed for this study. If this is the case, more information about the characteristic should be provided in the text , or at least as supplementary material

Authors: Thank you for your comments. We have included more information on the rTLsDeep package in the supplementary material.

Introduction

-          Lines 115-116: given that the target reader of Remote Sensing may not be familiar with deep learning methods, and taken into account that only CNNs are used, I suggest to include somewhere in this section a brief description (3-4 lines ad maximum) of what a convolutional neural network is and how it works

Authors: Implemented. Thanks for the suggestion. We have added:

“ CNN's algorithm takes in an input image (e.g. acquired from UAV), assigns importance to various aspects in the image (e.g. objects, like trees), and is able to differentiate one from the other (e.g. tree species) [36]”

Material and methods

-          Line 155 and figure 1. How was the timber damage classification shown here carried out? In figure’s legend it indicates that the map was produced by Georgia and Florida forest services, but it is not clear the subjacent basis. By means of field survey, remote sensing, direct assignation by the intensity of the hurricane?

Authors: Thank you for your comments. The timber damage classification map was produced by the Georgia Forestry Commission and Florida Forest Service. We believe they used a combination of intensity of the hurricane (wind speed data from NOAA (National Oceanic and Atmospheric Administration)) and FIA data to generate the map. The map is produced with coarse resolution and we used it herein only to illustrate the impact of Michael in the study area.

https://gatrees.org/wp-content/uploads/2020/01/Hurricane-MichaelTimber-Impact-Assessment-Georgia-October-10-11-2018-2.pdf

-          Line 182: Which are the characteristics of these three sites? Level of damage? ISs there any data on the pre-hurricane attributes of the forest in each site (age. Mean height, basal area, stand density)? How were these sites selected?

Authors: Thank you for your comments. The sites were pre-selected after the disturbance event based on visual assessment of pre- and post-disturbance Google Earth aerial images. In the field, we visited the pre-selected areas and selected the final sites covering the entire range of damage severity (light, moderate, severe and catastrophic; see Fig 1). As we were focused on individual tree level damage assessment from TLS, we didn’t assess any stand level attribute (e.g basal area, mean height, tree density)

-          Line 182: How was selected the location of the 4 plots within each site? Were plot homogeneous within and between sites, or they aimed to cover different degree of initial conditions or damage severity?

Authors: Thank you for your question. The plots were selected in order to cover all range of damage severity

-          Line 187-188: The authors should indicate which was the criteria for selecting the individual trees finally segmented from the cloud point. If only easy to see, free of competition trees, are subjectively selected, this could lead to some bias in the accuracy of the classification method that at least should be discussed in deeper detail

Authors: Thanks for pointing this out. We randomly selected the trees within our plots, especially to avoid introducing bias.

-          Line 189-190: This damage severity classification was specifically proposed for this work or it's been used previously ?In that case provide a reference

Authors: We developed this damage severity classification based on the previou study published by Rutedge et al. 2021. We have cited it in the figure caption now.

Rutledge, B.T., Cannon, J.B., McIntyre, R.K., Holland, A.M., Jack, S.B., 2021. Tree, stand, and landscape factors contributing to hurricane damage in a coastal plain forest: Post-hurricane assessment in a longleaf pine landscape. For. Ecol. Manag. 481, 118724 https://doi.org/10.1016/j.foreco.2020.118724

-          Line 202-204: change the numeration of the figures (currently figure 4 is presented in the text before figure 3)

Authors: Fixed. Thank you

-          Lines 214-216: I miss a short description of the characteristics of the six different CNN methods evaluated, focusing on their characteristics, and the difference between them (including topics as free access, easiness for using …). This information could be provided as a table, even as an appendix. Moreover, indicate why these methods were selected, and even, why among the deep learning methods, CNN were selected among others

Authors: Thank you for pointing this out. We have included a brief description of the CNN methods now.

-          Line 230: please provide description (formulae) of OD, F1-score and Kappa index

Authors: Implemented. Thanks

Results

-          Line 264: what do the authors mean by “the confussion matrices were symmetric”? Confussion matrices are not symmetric at all.

Authors: Thank you for pointing this out. We removed this sentence.

Discussion

-          Lines 318-320: This pre-training process has not been presented neither in methods section nor in results. Since this may be a real cause for the superiority of some CNN techniques over others this should be detailed in these sections.

Authors: Thank you for your comments. The pre-training process, also known as transfer learning, was used in all models except the simple CNN architecture. This process consists of using the models initially trained with the ImageNet database (Deng et al., 2009) and usually provide better results than training the network from scratch. We included a sentence regarding transfer learning in the methods section of the revised manuscript.

Deng, J., Dong, W., Socher, R., Li, L. J., Li, K., & Fei-Fei, L. (2009, June). Imagenet: A large-scale hierarchical image database. In 2009 IEEE conference on computer vision and pattern recognition (pp. 248-255). IEEE.

-          Lines 347-350: without demerit at all of the quality and interest of the work I consider that the authors should be more realistic in indicating and discussing here that there is still a long path to run until the proposed technique may be used at an operational scale 

Authors: Thanks. We have re-phase the sentence:

This is the first attempt to combine TLS and deep learning for classifying damage severity at the tree level. Despite the promising results found herein, there is still a long path to run until the proposed method may be applied at an operational scale. Improvements not only on damage severity classification, but also on efficient methods for individual tree extraction from TLS data are needed. We hope that the promising results and the open-source rTLsDeep R package developed in this study for classifying post-hurricane damage severity at the individual tree level will stimulate further research and applications not just in longleaf pine but other forest types in hurricane-prone regions.

Reviewer 3 Report

This is the review of the paper titled "Post-hurricane damage severity classification at the individual tree level using terrestrial laser scanning and deep learning". The authors manually segment individual tree to make dataset based on TLS data. Six CNNs (Densenet201, EfficientNet_b7, Inception_v3, Res-net152v2, VGG16, and a simple CNN) were compared to evaluate the impact of hurricanes on individual tree. The research has a certain amount of work. The use of language is good. However, there are some substantial problems with the submitted manuscript, so I propose rejection.

1 In the introduction, the authors first describe the hazards of hurricanes on forests, then assess the possibilities and shortcomings of TLS for hurricane hazards, and finally draw out the shortcomings of classifying images by exploring existing research on the identification of tree species based on CNNs. However, there is little description of how existing studies assessed the impact of hurricanes on forests. The introduction section is not very logical. The problems present do not correlate well with the references listed (line 115 - line 132).

2 The authors divided the individual tree after the impact of the hurricane into six categories (C1, C2, C3, C4, C5, C6). Do the six categories already available include all trees affected? Is there a reference for such a classification? If so, please add.

 3 The method proposed in this paper has some limitations. To my knowledge, existing individual tree segmentation methods based on TLS point clouds are still challenging for segmenting fallen trees (i.e., C3, C4 and C6). This point can also be reflected in the manuscript. Ninety trees are manually segmented (line 187). However, there is no guarantee of the accuracy of such individual tree segmentation using existing automatic methods. I propose to add experiments and discussions on the impact of the accuracy of individual tree segmentation on the ability of CNNs to predict hurricane impact.

 4 The experimental design is clear and well described. However, the description of six CNNs is poor. In addition, the parameter setting involved in six CNNs is not mentioned in the method. It is recommended to add a brief description of the six CNNs and their application (via diagrams and text) to the method.

 5 In the results section, please specify the exact values for the comparison of six CNNs. (i.e., line 247 – line 252)

 All of these make me believe that the manuscript should not be considered for publication.

Author Response

This is the review of the paper titled "Post-hurricane damage severity classification at the individual tree level using terrestrial laser scanning and deep learning". The authors manually segment individual tree to make dataset based on TLS data. Six CNNs (Densenet201, EfficientNet_b7, Inception_v3, Res-net152v2, VGG16, and a simple CNN) were compared to evaluate the impact of hurricanes on individual tree. The research has a certain amount of work. The use of language is good. However, there are some substantial problems with the submitted manuscript, so I propose rejection.

Authors: Thank you for your comments. While we don’t agree with your suggestion, we respect your decision. As described in the manuscript, the goal of this paper was not for developing or testing an algorithm for individual tree segmentation, but for developing a framework for classifying post-hurricane damage severity at tree level using deep learning architectures. Several studies using TLS focused on classifying species and tree-level attributes using deep learning or other methods have also followed the same approach of extracting the trees manually, as we want to produce the best dataset and address the questions related only to our proposed method. For operationalization, we agree that the individual tree extraction procedure would need to be automatized, but this is not the case for this study. 

1 In the introduction, the authors first describe the hazards of hurricanes on forests, then assess the possibilities and shortcomings of TLS for hurricane hazards, and finally draw out the shortcomings of classifying images by exploring existing research on the identification of tree species based on CNNs. However, there is little description of how existing studies assessed the impact of hurricanes on forests. The introduction section is not very logical. The problems present do not correlate well with the references listed (line 115 - line 132).

Authors: Thanks for pointing this out. We used the inverse cone strategy for writing the introduction and it is divided in 4 parts. First, a broad statement of the importance of southern pine forests and hazards of hurricanes is provided; second, we stated that remote sensing has been used for mapping forest attributes, but not for classifying post-hurricane at tree level using TLS. We emphasized that traditional methods for assessing post-hurricane conditions are limited by cost. Third, we introduced CNNs with the hypothesis that combining TLS and CNNs we could develop an efficient method for classifying post-hurricane damage severity at the tree level.

2 The authors divided the individual tree after the impact of the hurricane into six categories (C1, C2, C3, C4, C5, C6). Do the six categories already available include all trees affected? Is there a reference for such a classification? If so, please add.

Authors: Thanks for the questions. The six categories cover all ranges of possible damages ( from light to catastrophic). We developed this damage severity classification based on the previou study published by  (e.g. Rutedge et al. 2021). We have cited it in the figure caption now.

Rutledge, B.T., Cannon, J.B., McIntyre, R.K., Holland, A.M., Jack, S.B., 2021. Tree, stand, and landscape factors contributing to hurricane damage in a coastal plain forest: Post-hurricane assessment in a longleaf pine landscape. For. Ecol. Manag. 481, 118724 https://doi.org/10.1016/j.foreco.2020.118724

 3 The method proposed in this paper has some limitations. To my knowledge, existing individual tree segmentation methods based on TLS point clouds are still challenging for segmenting fallen trees (i.e., C3, C4 and C6). This point can also be reflected in the manuscript. Ninety trees are manually segmented (line 187). However, there is no guarantee of the accuracy of such individual tree segmentation using existing automatic methods. I propose to add experiments and discussions on the impact of the accuracy of individual tree segmentation on the ability of CNNs to predict hurricane impact.

Authors: Thank you for your comments. Again, as described in the manuscripts, the goal of this paper was not for developing or testing an algorithm for individual tree segmentation, but for developing a framework for classifying post-hurricane damage severity at tree level using deep learning architectures. Several studies using TLS focused on classifying species and tree-level attributes using deep learning or other methods have also followed the same approach of extracting the trees manually, as we want to produce the best dataset and address the questions related to deep learning and TLS, not single tree extraction. For operationalization, we agree that the individual tree extraction procedure would need to be automatized, but this is not the case for this study.  We have re-frame part of the text to address this issue. In specific, we added:

“This is the first attempt to combine TLS and deep learning for classifying damage severity at the tree level. Despite the promising results found herein, there is still a long path to run until the proposed method may be applied at an operational scale. Improvements not only on damage severity classification, but also on efficient and automatic methods for individual tree extraction, especially fallen trees, from TLS data are needed. We hope that the promising results and the open-source rTLsDeep R package developed in this study for classifying post-hurricane damage severity at the individual tree level will stimulate further research and applications not just in longleaf pine but other forest types in hurricane-prone regions. “

 4 The experimental design is clear and well described. However, the description of six CNNs is poor. In addition, the parameter setting involved in six CNNs is not mentioned in the method. It is recommended to add a brief description of the six CNNs and their application (via diagrams and text) to the method.

Authors: Thank you for pointing this out. We have added a brief description for the six CNNs methods now.

  5 In the results section, please specify the exact values for the comparison of six CNNs. (i.e., line 247 – line 252)

Authors:  Thanks for the suggestion. We have 72 results of F1-score and it is too much to include all in the text. We have created Figure 5, and all F1-score results are presented in that fig.

 All of these make me believe that the manuscript should not be considered for publication.

Authors: Thank you for your comments. While we don’t agree with your suggestion, we respect your decision. As part of this paper we have developed a new framework for classifying damage severity at the tree level using TLS data and we develop a new and open-source R package.

Round 2

Reviewer 1 Report

The authors have done all required corrections.